# Stochastic Concept Bottleneck Models

**Moritz Vandenhirtz,**[*] **Sonia Laguna,**[*] **Ričards Marcinkevičs, Julia E. Vogt**
Department of Computer Science
ETH Zurich
Switzerland

## Abstract

Concept Bottleneck Models (CBMs) have emerged as a promising interpretable method whose final prediction is based on intermediate, human-understandable concepts rather than the raw input. Through time-consuming manual interventions, a user can correct wrongly predicted concept values to enhance the model's downstream performance. We propose *Stochastic Concept Bottleneck Models* (SCBMs), a novel approach that models concept dependencies. In SCBMs, a single-concept intervention affects all correlated concepts, thereby improving intervention effectiveness. Unlike previous approaches that model the concept relations via an autoregressive structure, we introduce an explicit, distributional parameterization that allows SCBMs to retain the CBMs' efficient training and inference procedure. Additionally, we leverage the parameterization to derive an effective intervention strategy based on the confidence region. We show empirically on synthetic tabular and natural image datasets that our approach improves intervention effectiveness significantly. Notably, we showcase the versatility and usability of SCBMs by examining a setting with CLIP-inferred concepts, alleviating the need for manual concept annotations.

## 1 Introduction

In today's world, machine learning plays a crucial role in making important decisions, from healthcare to finance and law. However, as these algorithms become more complex, understanding how they arrive at their decisions becomes increasingly challenging. This lack of interpretability is a significant concern, especially in situations where trustworthiness, transparency, and accountability are paramount (Lipton, 2016; Doshi-Velez & Kim, 2017). Recent studies have focused on Concept Bottleneck Models (CBMs) (Koh et al., 2020; Havasi et al., 2022; Shin et al., 2023), a class of models that predict human-understandable concepts upon which the final target prediction is based. CBMs offer interpretability since a user can inspect the predicted concept values to understand how the model arrives at its final target prediction. Moreover, if they disagree with a concept prediction, they can intervene by adjusting it to the right value, which in turn affects the target prediction.

For example, consider the yellow warbler in Figure 1 (a), where a user might notice that the binary concept 'yellow primary color' is mispredicted. Upon this realization, they can intervene on the CBM by setting its value to 1, which increases the probability of the class yellow warbler. This way of interacting allows any untrained user to engage with the model to increase its predictive performance.

However, if the user input is that the primary color is yellow, should not the likelihood of a yellow crown increase too? This adaptation would increase the predicted likelihood of the correct class even more, as yellow warblers are characterized by their fully yellow body. Currently, vanilla CBMs do not exhibit this behavior as they do not use the intervened-on concepts to update their remaining concept predictions. This indicates that they suboptimally adapt to the additional knowledge gained.

---

[*]Equal contribution. Correspondence to `{moritz.vandenhirtz,slaguna}@inf.ethz.ch`

38th Conference on Neural Information Processing Systems (NeurIPS 2024).

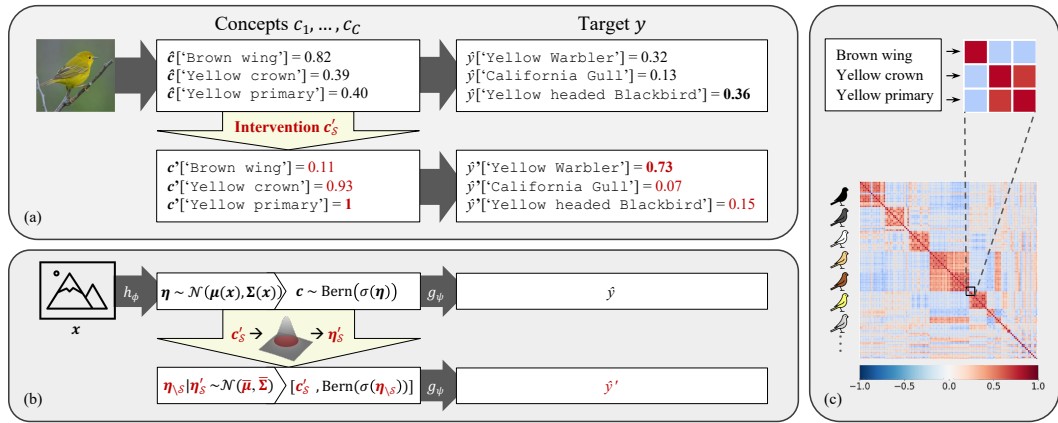

Figure 1: Overview of the proposed method for the CUB dataset. (a) A user intervenes on the concept of 'primary color: yellow'. Unlike CBMs, our method then uses this information to adjust the predicted probability of correlated concepts, thereby affecting the target prediction. (b) Schematic overview of the intervention procedure. A user's intervention $c'_{\mathcal{S}}$ is used to infer the logits $\boldsymbol{\eta}_{\backslash\mathcal{S}}$ of the remaining concepts. (c) Visualization of the learned global dependency structure as a correlation matrix for the 112 concepts of CUB (Wah et al., 2011). Characterization of concepts on the left.

To this end, we propose to extend the concept predictions with the modeling of their dependencies, as depicted in Figure 1.

The proposed approach captures the concept dependencies by modeling the concept logits with a learnable non-diagonal normal distribution, which enables efficient, scalable computing of the effect of interventions on other concepts. By integrating concept correlations, we reduce the time and effort of having to laboriously intervene on many correlated variables and increase the efficacy of interventions on the downstream prediction. Thanks to the explicit distributional assumptions, the model is trained end-to-end, retaining the training and inference speed of classic CBMs as well as the benefits of training the concept and target predictor jointly. Moreover, we show that our method excels when querying user interventions based on predicted concept uncertainty (Shin et al., 2023), further highlighting the practical utility of our approach as such policies spare users from manually sifting through the concepts to identify necessary interventions. Lastly, based on the distributional concept parameterization, we propose a novel approach for computing dependency-aware interventions through the likelihood-based confidence region.

**Contributions** This work contributes to the line of research on concept bottleneck models in several ways. (*i*) We propose to capture and model concept dependencies with a multivariate normal distribution. (*ii*) We derive a novel intervention strategy based on the confidence region of the normal distribution that incorporates concept correlations. Using the learned concept dependencies during the intervention procedure allows for stronger interventional effectiveness. (*iii*) We provide a thorough empirical assessment of the proposed method on synthetic tabular and natural image data. Additionally, we combine our method with concept discovery where we alleviate the need for annotations by using CLIP-inferred concepts. In particular, we show the proposed method (a) discovers meaningful, interpretable patterns in the form of concept dependencies, (b) allows for fast, scalable inference, and (c) outperforms related work with respect to intervention effectiveness thanks to the proposed concept modeling and intervention strategy.

## 2 Background & Related Work

Concept bottleneck models (Koh et al., 2020; Lampert et al., 2009; N. Kumar et al., 2009) are typically trained on data points $(\boldsymbol{x}, \boldsymbol{c}, y)$, comprising the covariates $\boldsymbol{x} \in \mathcal{X}$, target $y \in \mathcal{Y}$, and $C$ annotated binary concepts $\boldsymbol{c} \in \mathcal{C}$. Consider a neural network $f_{\boldsymbol{\theta}}$ parameterized by $\boldsymbol{\theta}$ and a slice $\langle g_{\psi}, h_{\phi} \rangle$ (Leino et al., 2018) s.t. $\hat{y} := f_{\boldsymbol{\theta}}(\boldsymbol{x}) = g_{\psi}(h_{\phi}(\boldsymbol{x}))$. CBMs enforce a concept bottleneck $\hat{\boldsymbol{c}} := h_{\phi}(\boldsymbol{x})$ such that the model's final output depends on the covariates $\boldsymbol{x}$ solely through the predicted concepts $\hat{\boldsymbol{c}}$.

While Koh et al. (2020) propose the *soft* CBM, where the concept logits parameterize the bottleneck, Havasi et al. (2022) argue that such a representation leads to leakage, where additional unwanted information in the concept representation is used to predict the target (Margeloiu et al., 2021; Mahinpei et al., 2021). Thus, they parameterize the bottleneck by binarized concept predictions and call it the *hard* CBM. Then, Havasi et al. (2022) equip the hard CBM with an autoregressive structure of the form $c_i|\boldsymbol{x}, \boldsymbol{c}_{<i}$, which is supposed to learn the concept dependencies. As such, the implicit autoregressive modeling of concept dependencies by Havasi et al. (2022) is the most related to the current work. Complementary to our work, Heidemann et al. (2023) analyze how a CBM's performance is affected by concept correlations. Unlike approaches that restrict the bottleneck to prevent leakage, Concept Embedding Models (CEM) (Espinosa Zarlenga et al., 2022) represent each concept with an embedding vector from which the concept probabilities can be inferred. E. Kim et al. (2023) model the embedding with a normal distribution, assuming a diagonal covariance matrix, which prevents them from capturing concept dependencies. Therefore, their intervention performance is not expected to differ from that of CEMs. Recent works explored how a CBM-like structure can be enforced even without a concept-annotated training set. Yuksekgonul et al. (2023) transform a pre-trained model into a CBM via a concept bank from concept activation vectors and multimodal models (B. Kim et al., 2018), while Oikarinen et al. (2023) query GPT-3 (Brown et al., 2020) for the concept set $\mathcal{C}$ and assign the values of the concept activations to each datapoint $\boldsymbol{x}$ with CLIP (Radford et al., 2021) similarities. Similarly, Panousis et al. (2023) uses CLIP to probabilistically discover a sparse set of concepts for each input, which could be used in our model for a fully probabilistic pipeline. Lastly, Marcinkevičs et al. (2024) instead relax the need for a concept labeled training set to a smaller validation set by fine-tuning a pre-trained model.

Intervenability (Marcinkevičs et al., 2024) is a crucial element of CBMs as it allows the user to correct wrongly predicted concepts $\hat{\boldsymbol{c}}$ to $\boldsymbol{c}'$, which in turn affects the target prediction of the model $\hat{y}'$. If multiple concepts are intervened on sequentially, the order of interventions is important. To this end, Sheth et al. (2022) and Shin et al. (2023) explore multiple policies according to which the order of concepts is determined. Chauhan et al. (2023) propose to combine predefined policies with learnable weighting parameters, while Espinosa Zarlenga et al. (2024) learn the policy itself. Concurrently, Singhi et al. (2024) learn a realignment module to align concept predictions. Steinmann et al. (2023) argue that instance-specific interventions are costly and store previous interventions in a memory to automatically reapply them for similar data points. Lastly, Collins et al. (2023) explore the advantages of including uncertainty rather than treating humans as oracles.

Our work models concept dependencies by parameterizing the bottleneck with a distribution. In a similar vein, Variational Autoencoders (Kingma & Welling, 2014) parameterize the bottleneck with a normal distribution to model and generate new data. Stochastic Segmentation Networks (Monteiro et al., 2020) parameterize the logits of a segmentation map with a non-diagonal normal distribution to capture the spatial correlations of pixels and model the aleatoric uncertainty. The modeling of uncertainty with a distribution is also explored by Bayesian Neural Networks (Neal, 1995) that learn a probability distribution over the neurons of a neural network.

## 3 Methods

We propose Stochastic Concept Bottleneck Models[1] (SCBM), a novel concept-based method that relaxes the implicit CBM assumption of independent concepts. SCBM captures the concept dependencies by learning their multivariate distribution. As a result, interventions become more effective and scalable, as a single intervention can influence multiple correlated concepts. A schematic overview of the proposed method is depicted in Figure 1 (b).

### 3.1 Model Formulation

To capture the concept dependencies, we model the concept logits $\boldsymbol{\eta}$ with a learned multivariate normal distribution. Modeling logits with a normal distribution has proven to be effective in the context of segmentation (Monteiro et al., 2020). While Monteiro et al. (2020) use it to capture the spatial dependencies of pixels, we, instead, model the relations between concepts, where the properties of the normal distribution will prove useful. A neural network is trained to predict the distribution's parameters $\boldsymbol{\eta} \mid \boldsymbol{x} \sim \mathcal{N}(\boldsymbol{\mu}(\boldsymbol{x}), \boldsymbol{\Sigma}(\boldsymbol{x}))$, where $\boldsymbol{\mu}(\boldsymbol{x}) \in \mathbb{R}^C$, and $\boldsymbol{\Sigma}(\boldsymbol{x}) \in \mathbb{R}^{C \times C}$. Thus, the traditional

---

[1]The code is available here: `https://github.com/mvandenhi/SCBM`.

assumption of independent concepts $c_i \perp\!\!\!\perp c_j \mid \boldsymbol{x}, \ \forall i \neq j$ is relaxed to $c_i \perp\!\!\!\perp c_j \mid \boldsymbol{\eta}, \ \forall i \neq j$, where the assumed normal distribution induces linear concept dependencies. The inductive bias of linearity is useful in practice as it is more robust to overfitting and computationally more scalable with respect to $C$ compared to its nonlinear alternative (Havasi et al., 2022), as we will show in Section 5.

To learn the distribution, we minimize the negative log-likelihood

$$- \log p(\boldsymbol{c} \mid \boldsymbol{x}) = - \log \int p(\boldsymbol{c} \mid \boldsymbol{\eta}) p_\phi(\boldsymbol{\eta} \mid \boldsymbol{x}) d\boldsymbol{\eta}, \tag{1}$$

where $\phi$ are the parameters of a neural network that predicts the distribution $\boldsymbol{\eta} \mid \boldsymbol{x} \sim \mathcal{N}(\boldsymbol{\mu}(\boldsymbol{x}), \boldsymbol{\Sigma}(\boldsymbol{x}))$. This integral is intractable due to the softmax operation applied in $p(\boldsymbol{c} \mid \boldsymbol{\eta})$. Thus, the integral is approximated by $M$ Monte Carlo samples

$$- \log \int p(\boldsymbol{c} \mid \boldsymbol{\eta}) p_\phi(\boldsymbol{\eta} \mid \boldsymbol{x}) d\boldsymbol{\eta} \approx - \log \frac{1}{M} \sum_{m=1}^{M} p(\boldsymbol{c} \mid \boldsymbol{\eta}^{(m)}), \quad \boldsymbol{\eta}^{(m)} \mid \boldsymbol{x} \sim \mathcal{N}(\boldsymbol{\mu}(\boldsymbol{x}), \boldsymbol{\Sigma}(\boldsymbol{x})). \tag{2}$$

In order to learn $\phi$, we make use of the parameterization as normal distribution and employ the reparameterization trick $\boldsymbol{\eta}^{(m)} \mid \boldsymbol{x} = \boldsymbol{\mu}(\boldsymbol{x}) + \mathbf{L}(\boldsymbol{x})\boldsymbol{\epsilon}^{(m)}, \quad \mathbf{L}(\boldsymbol{x})\mathbf{L}(\boldsymbol{x})^T = \boldsymbol{\Sigma}(\boldsymbol{x}), \quad \boldsymbol{\epsilon}^{(m)} \sim \mathcal{N}(\boldsymbol{0}, \boldsymbol{I})$ such that gradients can be computed with respect to the parameters. Lastly, we incorporate the new relaxed conditional independence assumption

$$\log p(\boldsymbol{c} \mid \boldsymbol{\eta}) = \log \prod_{i=1}^{C} p(c_i \mid \eta_i) = \sum_{i=1}^{C} \log p(c_i \mid \eta_i), \tag{3}$$

where $p(c_i \mid \eta_i)$ describes a Bernoulli distribution parameterized by the sigmoid-transformed logits $\sigma(\eta_i)$. Combining the above considerations results in the following reformulation of the negative log-likelihood:

$$
\begin{aligned}
- \log p(\boldsymbol{c} \mid \boldsymbol{x}) &\approx - \log \frac{1}{M} \sum_{m=1}^{M} p(\boldsymbol{c} \mid \boldsymbol{\eta}^{(m)}) \\
&\propto - \log \sum_{m=1}^{M} \exp \sum_{i=1}^{C} \log p(c_i \mid \eta_i^{(m)}) \\
&= - \log \sum_{m=1}^{M} \exp \sum_{i=1}^{C} \left[ -\mathrm{BCE}(c_i, \sigma(\eta_i^{(m)})) \right],
\end{aligned} \tag{4}
$$

where BCE stands for Binary Cross Entropy, and the $\mathrm{logsumexp}$ trick is used for numerical stability.

The distribution-based modeling procedure allows for efficient sampling, thus, enabling SCBM to train concept and target predictors jointly, sequentially, or independently. In contrast, the autoregressive alternative (Havasi et al., 2022) requires independent training due to the computational complexity. We adopt a joint training scheme to obtain the benefits of end-to-end learning where concept and target predictors can adjust to each other. To prevent leakage, we follow Havasi et al. (2022) and train the model with the hard $\{0, 1\}$ concept values as bottleneck rather than the logits used in the original CBM (Koh et al., 2020). To this end, we employ the straight-through Gumbel-Softmax trick (Jang et al., 2017; Maddison et al., 2017) that approximates Bernoulli samples while being differentiable. The target predictor $g_\psi$ is then learned by minimizing the negative log-likelihood

$$
\begin{aligned}
- \log p(y \mid \boldsymbol{x}) &= - \log \sum_{\boldsymbol{c} \in \mathcal{C}} p_\psi(y \mid \boldsymbol{c}) p(\boldsymbol{c} \mid \boldsymbol{x}) \\
&\approx - \log \frac{1}{M} \sum_{m=1}^{M} p_\psi(y \mid \boldsymbol{c}^{(m)}), \qquad \boldsymbol{c}^{(m)} \sim p(\boldsymbol{c} \mid \boldsymbol{x}).
\end{aligned} \tag{5}
$$

Lastly, the learned dependencies are regularized by following Occam's razor and to prevent overfitting. We take inspiration from the Graphical Lasso (Friedman et al., 2008) and penalize the off-diagonal elements of the precision matrix $\boldsymbol{\Sigma}^{-1}$.

By combining concept, target, and precision loss with weighting factors $\lambda_1$ and $\lambda_2$, we arrive at the final loss function

$$- \log \sum_{m=1}^{M} \exp \sum_{i=1}^{C} -\mathrm{BCE}\left(c_i, \sigma(\eta_i^{(m)})\right) + \lambda_1 \mathrm{CE}\left(y, \frac{1}{M} \sum_{m=1}^{M} g_\psi(\boldsymbol{c}^{(m)})\right) + \lambda_2 \sum_{i \neq j} \boldsymbol{\Sigma}(\boldsymbol{x})_{i,j}^{-1}. \tag{6}$$

## 3.2 Covariance Learning

The introduced amortized covariance matrix $\Sigma(x)$ provides the flexibility to tailor its predicted concept dependencies to each data point, making it adaptable to many data-generating mechanisms. For example, in the commonly used CUB (Wah et al., 2011; Koh et al., 2020), it can learn the class-wise concept structure present in the dataset. The explicit dependency representation inferred by the learned covariance matrix is useful as it provides insights into the learned correlations among the concepts, which is important for understanding and interpreting the model behavior.

However, an amortized covariance matrix comes at the price of not being able to visualize and interpret a unified concept structure on a dataset level. Depending on the need of the application, such a global structure might be preferable. Thus, we propose a variation of SCBM, where the covariance matrix is not *amortized* ($\Sigma(x)$), but learned *globally* ($\Sigma$). An example of the global concept structure learned on CUB is shown in Figure 1 (c). This variation has the inductive bias of assuming a constant covariance matrix, whose utility depends on the underlying data-generating mechanism. We recommend using the more flexible, amortized version by default and only utilizing a global covariance if the strong assumption of fixed dependencies is reasonable. We will explore this empirically in more detail in Section 5.

## 3.3 Interventions

A distinguishing property of CBM-like methods is the user's capacity to correct wrongly predicted concepts, which in turn affects the target prediction (Marcinkevičs et al., 2024). For a big concept set, this intervention procedure can become quite laborious as a user has to inspect and manually intervene on each concept separately. SCBMs are designed to alleviate this need by utilizing the learned concept dependencies such that a single intervention affects all related concepts as modeled by the multivariate normal distribution.

The parameterization as a multivariate normal distribution allows for a quick, scalable intervention procedure. Given a set $\mathcal{S} \subset \{1, \ldots, C\}$ of concept interventions, the effect on the remaining concepts $c_{\backslash \mathcal{S}}$ is computed via their logits $\eta_{\backslash \mathcal{S}}$ by conditioning on the intervention logits $\eta'_{\mathcal{S}}$, utilizing the known properties of the normal distribution

$$
\begin{aligned}
\eta_{\backslash \mathcal{S}} \mid x, \eta'_{\mathcal{S}} &\sim \mathcal{N}\left(\bar{\mu}(x), \overline{\Sigma}(x)\right), \\
\bar{\mu} &= \mu_{\backslash \mathcal{S}} + \Sigma_{\backslash \mathcal{S}, \mathcal{S}} \Sigma_{\mathcal{S}, \mathcal{S}}^{-1}(\eta'_{\mathcal{S}} - \mu_{\mathcal{S}}), \\
\overline{\Sigma} &= \Sigma_{\backslash \mathcal{S}, \backslash \mathcal{S}} - \Sigma_{\backslash \mathcal{S}, \mathcal{S}} \Sigma_{\mathcal{S}, \mathcal{S}}^{-1} \Sigma_{\mathcal{S}, \backslash \mathcal{S}}.
\end{aligned}
\tag{7}
$$

In standard CBMs, an intervention affects only the concepts on which the user intervenes. As such, Koh et al. (2020) set $\eta'_i$ to the 5th percentile of the training distribution if $c_i = 0$ and the 95th percentile if $c_i = 1$. While this strategy is effective for SCBMs too, see Appendix C.5, the modeling of the concept dependencies warrants a more thorough analysis of the *intervention strategy*. We present two desiderata, which our intervention strategy should fulfill.

*i)* $p(c_i \mid \eta'_i) \geq p(c_i \mid \mu_i)$

The likelihood of the intervened-on concept $c_i$ should always increase after the intervention. If SCBMs used the same strategy as CBMs, it could happen that the initially predicted $\mu_i$ was more extreme than the selected training percentile. Then, the interventional shift $\eta'_i - \mu_i$ in Eq. 7 would point in the wrong direction. This would cause $\eta_{\backslash \mathcal{S}}$ to shift incorrectly.

*ii)* $|\eta'_i - \mu_i|$ *should not be "too large"*.

We posit that the interventional shift should stay within a reasonable range of values. Otherwise, the effect on $\eta_{\backslash \mathcal{S}}$ would be unreasonably large such that the predicted $\mu_{\backslash \mathcal{S}}$ would be completely disregarded.

To fulfill these desiderata, we take advantage of the explicit distributional representation: the likelihood-based confidence region of $\mu_i$ provides a natural way of specifying the region of possible $\eta'_{\mathcal{S}}$ that fulfill our desiderata. Informally, a confidence region captures the region of plausible values for a parameter of a distribution. Note that the confidence region takes concept dependencies into account when describing the area of possible $\eta'_{\mathcal{S}}$. To determine the specific point within this region, we search for the values $\eta'_{\mathcal{S}}$, which maximize the log-likelihood of the known, intervened-on concepts $c_{\mathcal{S}}$, implicitly focusing on concepts that the model predicts poorly:

$$\boldsymbol{\eta}'_{\mathcal{S}} = \arg\max_{\boldsymbol{\eta}_{\mathcal{S}}} \log p(\boldsymbol{c}_{\mathcal{S}} \mid \boldsymbol{\eta}_{\mathcal{S}})$$
$$\text{s.t.} - 2\left(\log p(\boldsymbol{\eta}_{\mathcal{S}} \mid \boldsymbol{\mu}_{\mathcal{S}}, \boldsymbol{\Sigma}_{\mathcal{S},\mathcal{S}}) - \log p(\boldsymbol{\mu}_{\mathcal{S}} \mid \boldsymbol{\mu}_{\mathcal{S}}, \boldsymbol{\Sigma}_{\mathcal{S},\mathcal{S}})\right) \leq \chi^2_{d,1-\alpha} \qquad (8)$$
$$\eta'_i - \mu_i \geq 0 \text{ if } c_i = 1, \quad \forall i \in \mathcal{S}$$
$$\eta'_i - \mu_i \leq 0 \text{ if } c_i = 0, \quad \forall i \in \mathcal{S},$$

where $d = |\mathcal{S}|$. The first inequality describes the confidence region. It is based on the logarithm of the likelihood ratio, which, after multiplying with $-2$, asymptotically follows a $\chi^2$ distribution (Silvey, 1975). The last two inequalities restrict the region to the desired direction. Note that $\boldsymbol{\eta}'_{\mathcal{S}}$ is computed to determine the conditional effect of the interventions on $\boldsymbol{\eta}_{\setminus\mathcal{S}}$ using Equation 7. When predicting $\hat{y}'$ under interventions, the logits $\boldsymbol{\eta}_{\setminus\mathcal{S}}$ are then used for sampling the binary concept values $\boldsymbol{c}_{\setminus\mathcal{S}}$ while the intervened-on concepts $\boldsymbol{c}'_{\mathcal{S}}$ are directly set to their known, binary value.

## 4 Experimental Setup

**Datasets and Evaluation**  We perform experiments on a variety of datasets to showcase the validity of our method. Inspired by Marcinkevičs et al. (2024), we introduce a synthetic tabular dataset with a data-generating mechanism that contains fixed concept dependencies we can regulate. In particular, the concept logits $\boldsymbol{\eta}$ are sampled from a randomly initialized positive definite covariance matrix and generate $\boldsymbol{x}$. Binary concept values $\boldsymbol{c}$ are inferred from $\boldsymbol{\eta}$ and generate the target $y$. We refer to Appendix A.1 for a more detailed description.

As a natural image classification benchmark, we evaluate on the Caltech-UCSD Birds-200-2011 dataset (Wah et al., 2011), comprised of bird photographs from 200 distinct classes. It includes 112 concepts, such as wing color and beak shape, shared across the same class instances as revised in the original CBM work (Koh et al., 2020). Additionally, we explore another natural image classification task on CIFAR-10 (Krizhevsky et al., 2009) with 10 classes. To mitigate the concept annotations requirement, the concepts are synthetically acquired in a similar fashion to the concept discovery literature. We adopt the 143 concept classes generated via GPT-3 (Brown et al., 2020) in prior work (Oikarinen et al., 2023). To obtain the binary concept values, we use the CLIP model (Radford et al., 2021) to compute the similarity between each instance of an image with the text embedding of a specific concept and compare it to the similarity of its negative counterpart, i.e. *not* the concept. Appendix A.2 contains further details about the natural image datasets.

To compare methods, we evaluate the model performance based on the concept and target accuracy. We compute test performance before and after intervening on an increasing number of concepts. The order of concepts in the intervention is determined by an uncertainty-based policy (Shin et al., 2023) that selects the concept whose predicted probability is closest to $0.5$. We also show results for a random policy in Appendix C.3. Additionally, we evaluate the calibration of the predicted concept uncertainties that are being used for the uncertainty-based policy, with the Brier score (Brier, 1950) and the Expected Calibration Error (Naeini et al., 2015; A. Kumar et al., 2019).

**Baselines**  We evaluate the performance of our method in comparison with state-of-the-art models. Namely, we focus on the vanilla concept bottleneck model (CBM) by Koh et al. (2020) in its *hard* version (Havasi et al., 2022), trained jointly using the straight-through Gumbel-Softmax trick (Jang et al., 2017; Maddison et al., 2017), as a sensical baseline to our binary modeling of concepts. Additionally, we explore the concept embedding model (CEM) by Espinosa Zarlenga et al. (2022) that learns two concept embeddings, $\hat{\boldsymbol{c}}_i^+$ and $\hat{\boldsymbol{c}}_i^-$. These representations are used to predict the final concept probability with a learnable scoring function $\hat{p}_i = s(\hat{\boldsymbol{c}}_i^+, \hat{\boldsymbol{c}}_i^-) = \sigma(\mathbf{W}_s[\hat{\boldsymbol{c}}_i^+, \hat{\boldsymbol{c}}_i^-]^T + \mathbf{b}_s)$ and are then combined into a final concept embedding $\hat{\boldsymbol{c}}_i = (\hat{p}_i\hat{\boldsymbol{c}}_i^+ + (1 - \hat{p}_i)\hat{\boldsymbol{c}}_i^-)$ that is passed to the target predictor. Interventions are modeled by altering the concept probabilities $\hat{p}_i$. Note that Espinosa Zarlenga et al. (2022) optimize for intervention performance during training, which we omit, to ensure a fair comparison where no method was explicitly trained for intervention performance. Finally, we evaluate the autoregressive CBM structure proposed by Havasi et al. (2022), where concept dependencies are learned with an autoregressive structure. Here, each concept $c_i$ is predicted with a separate MLP that takes as input a latent representation of the input $f_{\boldsymbol{\theta}}(\boldsymbol{x})$ and all previous concepts $c_1, ..., c_{i-1}$. To obtain a good initialization of the autoregressive structure, it is pretrained

Table 1: Test-set concept and target accuracy (%) prior to interventions. Results are reported as averages and standard deviations of model performance across ten seeds. For each dataset and metric, the best-performing method is **bolded** and the runner-up is underlined.

| Dataset | Method | Concept Accuracy | Target Accuracy |
|---|---|---|---|
| Synthetic | Hard CBM | $61.42 \pm 0.07$ | $58.38 \pm 0.39$ |
| | CEM | $61.42 \pm 0.12$ | $58.01 \pm 0.49$ |
| | Autoregressive CBM | $\underline{62.17} \pm 0.11$ | $\mathbf{59.60} \pm 0.62$ |
| | Global SCBM | $61.57 \pm 0.05$ | $58.39 \pm 0.53$ |
| | Amortized SCBM | $\mathbf{62.41} \pm 0.20$ | $\underline{58.96} \pm 0.38$ |
| CUB | Hard CBM | $94.97 \pm 0.07$ | $67.72 \pm 0.57$ |
| | CEM | $95.12 \pm 0.07$ | $\underline{69.60} \pm 0.30$ |
| | Autoregressive CBM | $\mathbf{95.33} \pm 0.07$ | $69.24 \pm 0.44$ |
| | Global SCBM | $94.99 \pm 0.09$ | $68.19 \pm 0.63$ |
| | Amortized SCBM | $\underline{95.22} \pm 0.09$ | $\mathbf{69.87} \pm 0.56$ |
| CIFAR-10 | Hard CBM | $85.51 \pm 0.04$ | $69.73 \pm 0.29$ |
| | CEM | $85.12 \pm 0.14$ | $\mathbf{72.24} \pm 0.33$ |
| | Autoregressive CBM | $85.31 \pm 0.06$ | $68.88 \pm 0.47$ |
| | Global SCBM | $\underline{85.86} \pm 0.04$ | $70.74 \pm 0.29$ |
| | Amortized SCBM | $\mathbf{86.00} \pm 0.03$ | $\underline{71.66} \pm 0.25$ |

for 50 epochs. As the Monte Carlo sampling from the autoregressive structure is time-consuming, the target predictor $g_\psi$ is trained independently using the ground-truth concepts as input. At intervention time, a normalized importance sampling algorithm is used to estimate the concept distribution.

**Implementation Details** The model architectures comprise a backbone for concept prediction followed by a linear layer as head for an interpretable target prediction. More details can be found in Appendix B. To ensure the positive definiteness of the concept covariance matrix $\mathbf{\Sigma}$, we parameterize it via its Cholesky decomposition $\mathbf{\Sigma} = \boldsymbol{L}\boldsymbol{L}^\top$. Thus, we directly predict the lower triangular Cholesky matrix $\boldsymbol{L}$. We will evaluate two options for SCBMs: using a *global* ($\mathbf{\Sigma}$) or an *amortized* covariance matrix ($\mathbf{\Sigma}(\boldsymbol{x})$). For the amortized version, we set the weighting terms $\lambda_1$ and $\lambda_2$ of Equation 6 to 1. For the global version, we initialize it with the estimated empirical covariance matrix and set $\lambda_2 = 0$, as we did not observe big differences when varying $\lambda_2$. In Appendix C.4, we provide an ablation study, demonstrating that SCBMs are not very sensitive to the choice of $\lambda_2$. At intervention time, we solve the optimization problem based on the 99%-confidence region with the SLSQP algorithm (Kraft, 1988). In Appendix C.6, we provide an ablation with different confidence levels.

## 5 Results

**Test performance** In Table 1, we report the results of the concept and target accuracy prior to interventions. Overall, SCBM performs on par with the baseline methods, with no clear outperforming or underperforming technique throughout the datasets. In Appendix C.7, we show that other metrics lead to the same interpretation. This shows that the additional overhead of learning the concept dependencies does not negatively affect the predictive performance. We note that the amortized covariance variant con-

Table 2: Relative time it takes for one epoch in the CUB dataset when training on the training set, or evaluating on the test set, respectively.

| Method | Training | Inference |
|---|---|---|
| Hard CBM | 5x | 1x |
| CEM | 5x | 1x |
| Autoregressive CBM | 5x | 15x |
| Global SCBM | 5x | 1x |
| Amortized SCBM | 5x | 1x |

sistently surpasses the globally learned matrix due to its ability to adjust the predicted concept dependency structure and uncertainty on an instance level. On the other hand, the global variant offers a unified understanding of the concept correlations, an example of which is presented in Figure 1 (c). Notably, in CIFAR-10, even though the concept performance of CEM is the worst of all methods, it has the best target performance. This might suggest the presence of leakage in CEM's embeddings, as in CIFAR-10, the concept set alone is not sufficient to predict the target, and learning

additional information might be useful. In Table 2, we show the time it takes for training and testing of the methods. It is evident that the autoregressive CBM of Havasi et al. (2022) suffers from a slow sampling process due to its autoregressive structure, while SCBMs retain the efficiency of CBMs.

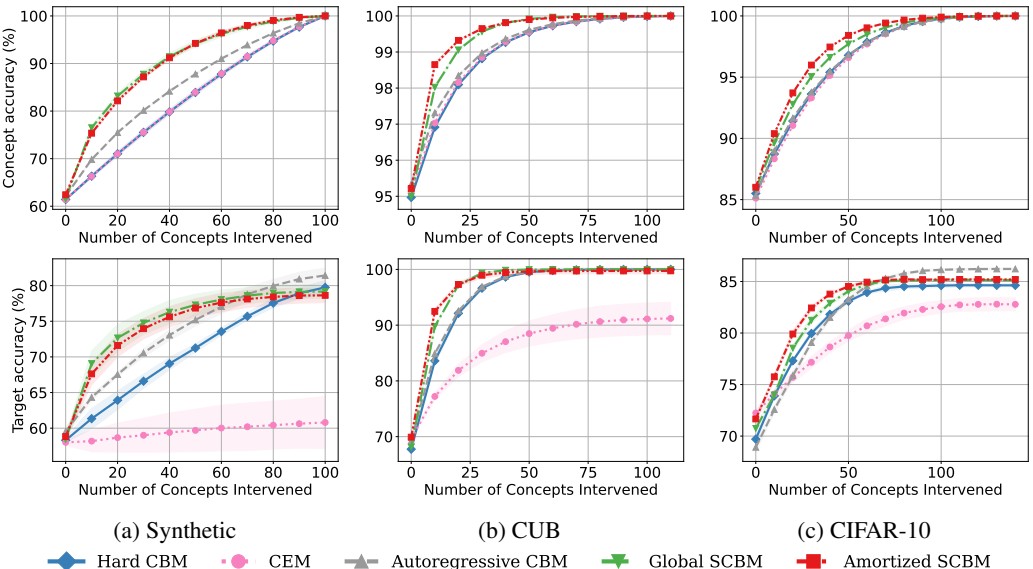

(a) Synthetic  (b) CUB  (c) CIFAR-10

Hard CBM ◆ CEM ● Autoregressive CBM ▲ Global SCBM ▼ Amortized SCBM ■

Figure 2: Performance after intervening on concepts in the order of highest predicted uncertainty. Concept and target accuracy (%) are shown in the first and second rows, respectively. Results are reported as averages and standard deviations of model performance across ten seeds.

**Interventions** In this paragraph, we analyze the intervention performance of SCBMs and their baseline models, focusing on their effectiveness in modeling concept dependencies and improving target accuracy. Figure 2 shows the intervention curves across ten seeds, where the performance is measured based on the concept and target accuracy. The order of concepts to intervene on is determined by an uncertainty-based policy that makes use of the predicted probabilities. In Appendix C.3, we present the intervention performance if concepts were selected randomly. The intervention curves in the first row show that SCBMs are superior in modeling the concept dependencies, as evidenced by their significantly steeper intervention curves compared to the baseline methods. Furthermore, the second row of Figure 2 indicates that the strong concept modeling translates to a significant improvement in downstream performance, partly thanks to the intervention strategy introduced in Section 3.3. We note that especially for the most practical scenario of only a small number of interventions, SCBMs outperform their counterparts. Comparing the SCBM variants, the natural image datasets show an overall better intervention performance with the amortized covariance matrix, following the trend of Table 1, as it can capture the instance-wise correlation structure of the data. Only in the synthetic dataset, where the data-generating covariance matrix is fixed, does the global SCBM slightly outperform the amortized one. Thus, we advocate for the usage of the global variant only if the underlying assumption of a fixed covariance is reasonable. Lastly, the success of SCBMs on CIFAR-10, with CLIP-based concepts, shows our proposed method can work without human-annotated concepts. To strengthen this point and also showcase the scalability of our method, in Appendix C.1, we provide results on CIFAR-100 with 892 concepts, where our SCBMs also strongly outperform baselines.

Analyzing the performance of the autoregressive CBM, which also captures concept dependencies, we observe that they expectedly have a better intervention performance than the hard vanilla CBM, which does not take correlations into account. However, it becomes evident that, compared to the concept performance of SCBMs, their autoregressive structure does not capture the dependencies to the full extent. This shows in the target accuracy, where they only match or outperform SCBMs towards the full set of intervened concepts. We attribute the better performance on the full intervention set to the independent training procedure utilized by autoregressive CBMs, which comes at the cost of lower test performance in CIFAR-10. Arguably, in a realistic use-case, such a high number of instance-level interventions is not sensible, and if it were, SCBMs could also be trained independently. Finally, the CEM shows reduced intervention performance as the expressive concept embeddings, which are prone to information leakage, seem to suboptimally adapt to the injected concept information.

Table 3: Test-set calibration (%) of concept predictions. Results are reported as averages and standard deviations of model performance across ten seeds. For each dataset and metric, the best-performing method is **bolded** and the runner-up is underlined. Lower is better.

| Dataset | Method | Brier | ECE |
|---|---|---|---|
| Synthetic | Hard CBM | $28.79 \pm 0.09$ | $22.38 \pm 0.15$ |
| | CEM | $29.32 \pm 0.08$ | $23.55 \pm 0.09$ |
| | Autoregressive CBM | $\mathbf{24.84} \pm 0.32$ | $\mathbf{13.54} \pm 0.49$ |
| | Global SCBM | $27.73 \pm 0.09$ | $20.10 \pm 0.14$ |
| | Amortized SCBM | $\underline{25.58} \pm 0.20$ | $\underline{15.57} \pm 0.55$ |
| CUB | Hard CBM | $3.93 \pm 0.05$ | $2.44 \pm 0.06$ |
| | CEM | $4.04 \pm 0.05$ | $3.25 \pm 0.07$ |
| | Autoregressive CBM | $\underline{3.75} \pm 0.05$ | $2.73 \pm 0.05$ |
| | Global SCBM | $3.87 \pm 0.06$ | $\underline{2.33} \pm 0.09$ |
| | Amortized SCBM | $\mathbf{3.64} \pm 0.07$ | $\mathbf{1.85} \pm 0.08$ |
| CIFAR-10 | Hard CBM | $10.42 \pm 0.05$ | $4.93 \pm 0.17$ |
| | CEM | $11.06 \pm 0.16$ | $7.11 \pm 0.39$ |
| | Autoregressive CBM | $10.70 \pm 0.05$ | $6.07 \pm 0.10$ |
| | Global SCBM | $\underline{9.95} \pm 0.02$ | $\underline{2.88} \pm 0.11$ |
| | Amortized SCBM | $\mathbf{9.84} \pm 0.02$ | $\mathbf{2.22} \pm 0.12$ |

Figure 3: Intervention performance of SCBMs measured in concept and target accuracy (%) on CUB for random and uncertainty-based policy.

**Modeling the concept distribution** A cornerstone of SCBMs is the explicit, distributional parameterization of concepts. This helps in understanding the data correlations and allows for visualization, as the example seen in Figure 1 (c). The explicit probabilistic modeling results in improved concept uncertainty estimates compared to the baseline CBM counterparts, as shown in Table 3, where lower metrics imply better estimates. This proves useful for interventions, where the uncertainty estimates can be leveraged for the choice of concepts to intervene on, improving the target prediction more effectively and reducing the need for manual user inspection. In Figure 3, we compare the performance of randomly intervening versus intervening based on the predicted uncertainty. We observe that there is a big gap between the two policies, indicating the usefulness of the estimated probabilities. Nevertheless, note that intervening at random remains successful and supports the observations made in the previous paragraph, as shown in Appendix C.3.

# 6 Conclusion

In this paper, we introduced SCBMs, a new concept-based method that models concept dependencies with a multivariate normal distribution. We proposed a novel, effective intervention strategy that takes concept correlations into account and is based on the confidence region inferred from the distributional parameterization. We showed that our modeling approach retains CBMs' training and inference speed, thus, being able to harness the benefits of end-to-end concept and target training. Additionally, the explicit parameterization offers the user a clearer understanding of the learned concept dependencies, providing deeper insights into how predictions and interventions are made. Empirically, we demonstrated that by modeling the concept dependencies, SCBMs offer a substantial improvement in intervention effectiveness, in concept as well as target accuracy, compared to related work. We showed that our method excels when iteratively intervening on the most uncertain concept predictions, sparing users from having to manually search through the concept set to identify necessary interventions. Additionally, our results indicate that learning the concept correlations does not decrease performance prior to interventions, in many cases even improving the performance over the baselines. Finally, the versatility of SCBMs is highlighted through their superior performance on CIFAR-10 and CIFAR-100, where concept values are CLIP-based rather than human-annotated.

**Limitations & Future Work**   This work opens multiple new research avenues. A natural extension is to go beyond binary concepts, such as continuous domains with their corresponding adaptations of modeling the concept distribution. Additionally, addressing the quadratic memory complexity of the covariance matrix is essential for scaling to larger concept sets. Our proposed intervention strategy accounts for model uncertainty, but further research is needed to accommodate user uncertainty, as human interventions are not always the ground truth. This work allows the editing of the learned dependency structure by adjusting the entries of the predicted covariance matrix, which could be explored. Lastly, to model additional information and reduce leakage, Koh et al. (2020); Havasi et al. (2022) propose the adoption of a side channel. The complementary effectiveness of incorporating the side channel in the covariance structure could be explored in the context of SCBMs.

## Acknowledgments and Disclosure of Funding

We thank Alexander Marx for the insightful discussions. MV and SL are supported by the Swiss State Secretariat for Education, Research, and Innovation (SERI) under contract number MB22.00047. RM is supported by the SNSF grant #320038189096.

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

# A  Dataset Details

In this section, we provide additional details on the datasets that are being used in the experiments.

## A.1  Synthetic Data-Generating Mechanism

Here, we describe the data-generating mechanism of the synthetic dataset in more detail. Let $N$, $p$, and $C$ denote the number of independent data points $\{(\boldsymbol{x}_n, \boldsymbol{c}_n, y_n)\}_{n=1}^{N}$, covariates, and concepts, respectively. We set $N = 50{,}000$, $p = 1{,}500$, and $C = 100$, with a 60%-20%-20% train-validation-test split. The generative process is as follows:

1. Randomly sample $\boldsymbol{W} \in \mathbb{R}^{C \times 10}$ s.t. $w_{i,j} \sim \mathcal{N}(0,1)$ for $1 \leq i \leq C$ and $1 \leq j \leq 10$.
2. Generate a positive definite matrix $\boldsymbol{\Sigma} \in \mathbb{R}^{C \times C}$ s.t. $\boldsymbol{\Sigma} = \boldsymbol{W}\boldsymbol{W}^T + \boldsymbol{D}$. Let $\boldsymbol{D} \in \mathbb{R}^{C \times C}$ s.t. $\boldsymbol{D} = \boldsymbol{\delta I}$, where $\delta_i \sim \mathcal{U}_{[0,1]}$ for $1 \leq i \leq C$.
3. Randomly sample logits $\boldsymbol{H} \in \mathbb{R}^{N \times C}$ s.t. $\boldsymbol{\eta}_n \sim \mathcal{N}(\boldsymbol{0}, \boldsymbol{\Sigma})$ for $1 \leq n \leq N$.
4. Let $c_{n,i} = \mathbb{1}_{\{\eta_{n,i} \geq 0\}}$ for $1 \leq n \leq N$ and $1 \leq i \leq C$.
5. Let $h : \mathbb{R}^C \to \mathbb{R}^p$ be a randomly initialised multilayer perceptron with ReLU nonlinearities.
6. Let $\boldsymbol{x}_n = h(\boldsymbol{\eta}_n) + \boldsymbol{\epsilon}_n$ s.t. $\boldsymbol{\epsilon}_n \sim \mathcal{N}(\boldsymbol{0}, \boldsymbol{I})$ for $1 \leq n \leq N$.
7. Let $g : \mathbb{R}^C \to \mathbb{R}$ be a randomly initialized linear perceptron.
8. Let $y_n = \mathbb{1}_{\{(g(\boldsymbol{c}_n) \geq y_{med})\}}$ for $1 \leq n \leq N$, where $y_{med}$ denotes the median of $g(\boldsymbol{c}_n)$.

## A.2  Natural Image Datasets

**Caltech-UCSD Birds-200-2011**   We evaluate on the Caltech-UCSD Birds-200-2011 (CUB)[2] dataset (Wah et al., 2011). It comprises 11,788 photographs from 200 distinct bird species annotated with 312 concepts, such as belly color and pattern. In this manuscript, we follow the original train-test split and revised the proposed dataset in the initial CBM work (Koh et al., 2020). Here, only the 112 most widespread binary attributes are included in the final dataset, and concepts are shared across samples in identical classes. The images were resized to a resolution of 224 × 224 pixels. Finally, following the original proposed augmentations, we applied random horizontal flips, modified the brightness and saturation, and applied normalization during training.

**CIFAR-10**   CIFAR-10[3] (Krizhevsky et al., 2009) is a natural image benchmark with 60,000 32x32 colour images and 10 classes. We kept the original train-test split, with 50,000 samples in the train set and a balanced total of 6,000 images per class. We generated 143 concept labels as described in Section 4 using large language and vision models. At training time, as for CUB, we applied augmentations including modifications to brightness and saturation, random horizontal flips and normalisation. Images were rescaled to a size of 224 × 224 pixels.

# B  Implementation Details

This section provides further implementation details of SCBM and the evaluated baselines. All methods were implemented using PyTorch (v 2.1.1) (Ansel et al., 2024). All models are trained for 150 epochs for the synthetic and 300 epochs for the natural image datasets with the Adam optimizer (Kingma & Ba, 2015) with a learning rate of $10^{-4}$ and a batch size of 64. For the independently trained autoregressive model, we split the training epochs into $2/3$ for the concept predictor and $1/3$ for the target predictor. For the methods requiring sampling, the number of Monte Carlo samples is set to $M = 100$. We provide an ablation for $M = 10$ in Appendix C.2. Note that since the predictor head is very simple, the MC sampling of SCBMs is extremely fast and does not influence computational complexity by more than $0.1\%$. For the synthetic tabular data, we use a fully connected neural network as backbone, with 3 non-linear layers, batch normalization, and dropout. For the CUB dataset, we use a pretrained ResNet-18 (He et al., 2016), and for the lower-resolution

---

[2]`https://www.vision.caltech.edu/datasets/cub_200_2011/`, no license available
[3]`https://www.cs.toronto.edu/~kriz/cifar.html`, no license available

CIFAR-10 a simple convolutional neural network with 2 convolutional layers followed by ReLU, Dropout, and a fully connected layer. For fairness in the comparisons, all baselines have the same model architecture choices and all experiments are performed over 10 random seeds.

**Resource Usage**   For the experiments of the main paper, we used a cluster of mostly GeForce RTX 2080s with 2 CPU workers. Over all methods, we estimate an average runtime of 8h per experiment, each running on a single GPU. This amounts to 5 methods × 3 datasets × 10 seeds × 8 hours = 1200 hours. Adding to that, the Ablation Figures required another 40 runs, amounting to a full total of 1520 hours of compute. Please note that we only report the numbers to generate the final results but not the development time, which we roughly estimate to be around 10 times bigger.

## C   Further Experiments

In this section, we show additional experiments to provide a more in-depth understanding of SCBM's effectiveness. We ablate multiple hyperparameters to provide an understanding of how they influence the model performance, as well as show the performance of our model in other settings.

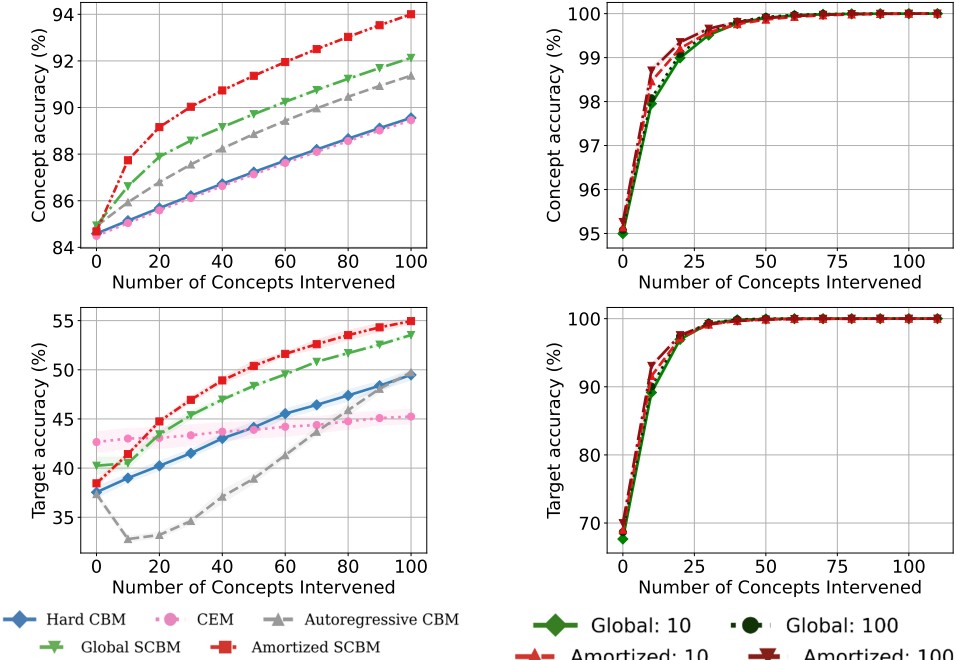

Figure 4: Performance after intervening on concepts in the order of highest predicted uncertainty in CIFAR-100 with 892 concepts. Concept and target accuracy (%) are shown in the first and second rows, respectively. Results are reported as averages and standard deviations of model performance across 3 seeds.

Figure 5: Intervention performance in the order of highest predicted uncertainty in CUB. Concept and target accuracy (%) are shown in the first and second rows, respectively. Results are reported as averages and standard deviations of model performance across 3 seeds.

### C.1   Intervention Performance on CIFAR-100

We present the result on the CIFAR-100 dataset with 892 concepts obtained from Oikarinen et al. (2023) in Figure 4 to showcase the scalability of SCBMs. The results underline the efficiency of our method. Notably, the Autoregressive baseline has a negative dip, which is likely due to the independently trained target predictor not being aligned with the concept predictors in this noisy CLIP-annotated scenario. Note that they need to train independently to avoid the sequential MC sampling during training, which would otherwise increase training time significantly. Our jointly

trained SCBMs do not have this issue and surpass the baselines. We use the same configuration as for CIFAR-10, with the exception that we set $M = 10$ to reduce the memory requirement.

## C.2   Number of Monte Carlo Samples

To showcase that SCBMs do not rely on a huge number of Monte Carlo samples, we provide an ablation of $M$ in Figure 5. It shows that even for $M = 10$, SCBMs thrive. Note, however, that since $M$ is not a driving factor of SCBMs computational cost, one can leave it at a high number.

## C.3   Random Intervention Policy

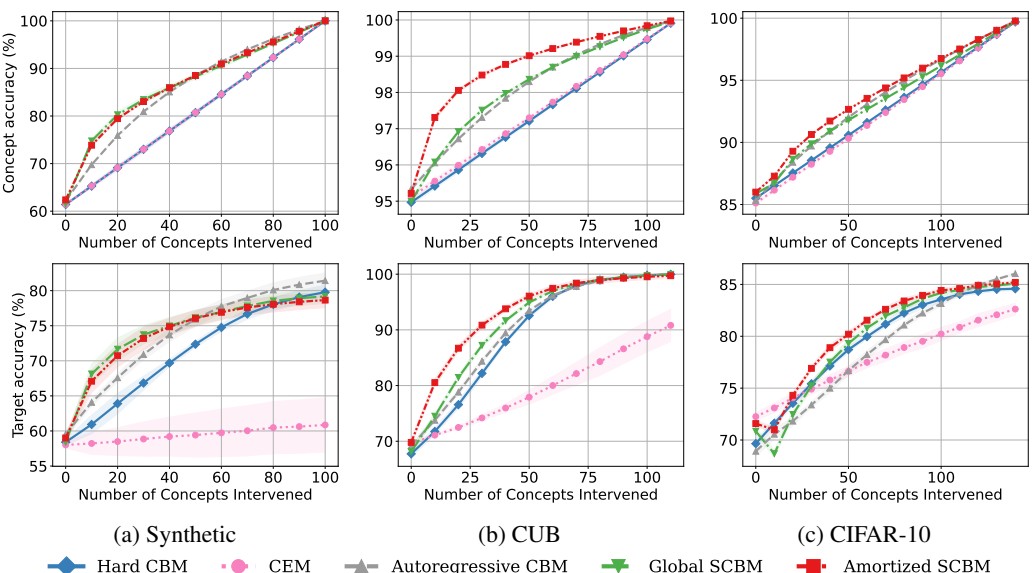

Figure 6: Performance after intervening on concepts in random order. Concept and target accuracy (%) are shown in the first and second rows, respectively. Results are reported as averages and standard deviations of model performance across ten seeds.

In Figure 6, we present the intervention performance of SCBM and baseline methods. Compared to the uncertainty-based intervention policy of Figure 2, the intervention curves of all methods are less steep, confirming the usefulness of Shin et al. (2023)'s proposed policy. Following the previous statements, SCBMs still outperform baseline methods with the amortized beating the global variant for real-world datasets. We observe that in CIFAR-10 for the first interventions, an improvement in concept accuracy is not directly reflected in improved target prediction for SCBMs, which is likely due to the low signal-to-noise ratio of the CLIP-inferred concepts.

## C.4   Regularization Strength

In Figure 7, we analyze the impact of the strength of $\lambda_2$ from Equation 6. Due to environmental considerations, we conducted experiments using only 5 seeds and limited the number of interventions to 20. Our findings indicate that SCBMs are not sensitive to the choice of $\lambda_2$, except that the unregularized amortized variant exhibits slight patterns of overfitting.

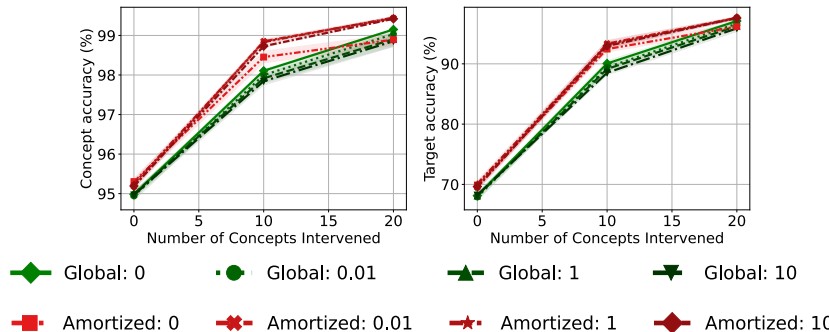

Figure 7: Performance on CUB after intervening on concepts in the order of highest predicted uncertainty with differing regularization strengths. Concept and target accuracy (%) are shown in the first and second columns, respectively. Results are reported as averages and standard deviations of model performance across five seeds. For each SCBM variant, we choose a darker color, the higher the regularization strength of $\lambda_2$.

## C.5 Intervention Strategy

In Figure 8, we analyze the effect of the intervention strategy. Our findings indicate that while SCBMs are still effective with the proposed strategy from Koh et al. (2020), that sets the logits to the 5th (if $c_i = 0$) or 95th (if $c_i = 1$) percentile of the training distribution, our proposed strategy based on the confidence region results in stronger intervenability.

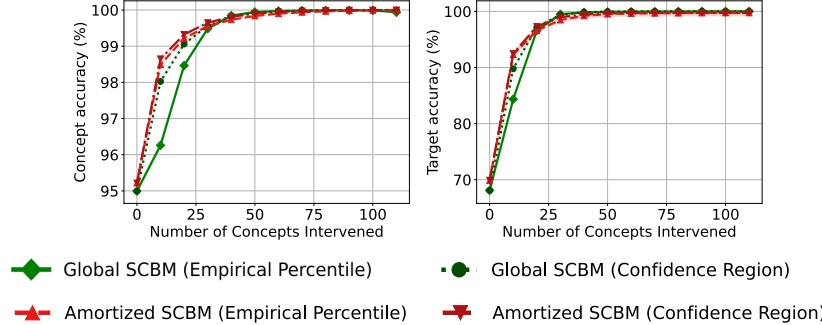

Figure 8: Performance on CUB after intervening on concepts in the order of highest predicted uncertainty, comparing the proposed intervention strategy to Koh et al. (2020)'s intervention of setting the logits to the 5th or 95th empirical percentile of the training distribution. Concept and target accuracy (%) are shown in the first and second columns, respectively. Results are reported as averages and standard deviations of model performance across five seeds.

## C.6 Confidence Region Level

In Figure 9, we analyze the effect of the level $1 - \alpha$ of the likelihood-based confidence region. Our findings indicate that the SCBMs are not sensitive to the choice of $1 - \alpha$, with higher levels being slightly better in performance.

## C.7 Jaccard Index

Panousis et al. (2024) propose to interpret the interpretation capacity of concepts with the Jaccard Index (Jaccard, 1901). As such, in Table 4, we extend Table 1 with this metric. It is evident that the interpretation does not change, indicating that the performance is robust to the choice of evaluation metric.

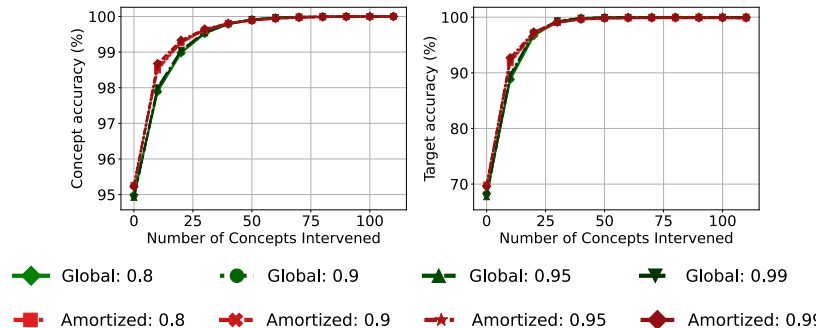

Figure 9: Performance on CUB after intervening on concepts in the order of highest predicted uncertainty with differing levels $1 - \alpha$ of the confidence region. Concept and target accuracy (%) are shown in the first and second columns, respectively. Results are reported as averages and standard deviations of model performance across three seeds.

Table 4: Test-set performance before interventions. Results are averaged across ten seeds.

| Dataset | Method | Concept Accuracy | Concept Jaccard | Target Accuracy |
|---------|--------|------------------|-----------------|-----------------|
| Synthetic | Hard CBM | $61.42 \pm 0.07$ | $43.80 \pm 1.32$ | $58.38 \pm 0.39$ |
| | CEM | $61.42 \pm 0.12$ | $44.84 \pm 1.36$ | $58.01 \pm 0.49$ |
| | Autoregressive CBM | $\underline{62.17} \pm 0.11$ | $\underline{45.30} \pm 1.29$ | $\mathbf{59.60} \pm 0.62$ |
| | Global SCBM | $61.57 \pm 0.05$ | $44.53 \pm 1.02$ | $58.39 \pm 0.53$ |
| | Amortized SCBM | $\mathbf{62.41} \pm 0.20$ | $\mathbf{45.85} \pm 1.45$ | $\underline{58.96} \pm 0.38$ |
| CUB | Hard CBM | $94.97 \pm 0.07$ | $77.22 \pm 0.33$ | $67.72 \pm 0.57$ |
| | CEM | $95.12 \pm 0.07$ | $78.20 \pm 0.28$ | $\underline{69.60} \pm 0.30$ |
| | Autoregressive CBM | $\mathbf{95.33} \pm 0.07$ | $\mathbf{79.21} \pm 0.21$ | $69.24 \pm 0.44$ |
| | Global SCBM | $94.99 \pm 0.09$ | $76.83 \pm 0.47$ | $68.19 \pm 0.63$ |
| | Amortized SCBM | $\underline{95.22} \pm 0.09$ | $\underline{78.29} \pm 0.28$ | $\mathbf{69.87} \pm 0.56$ |
| CIFAR-10 | Hard CBM | $85.51 \pm 0.04$ | $81.54 \pm 0.08$ | $69.73 \pm 0.29$ |
| | CEM | $85.12 \pm 0.14$ | $81.06 \pm 0.21$ | $\mathbf{72.24} \pm 0.33$ |
| | Autoregressive CBM | $85.31 \pm 0.06$ | $81.31 \pm 0.10$ | $68.88 \pm 0.47$ |
| | Global SCBM | $\underline{85.86} \pm 0.04$ | $\underline{81.81} \pm 0.19$ | $70.74 \pm 0.29$ |
| | Amortized SCBM | $\mathbf{86.00} \pm 0.03$ | $\mathbf{81.97} \pm 0.20$ | $\underline{71.66} \pm 0.25$ |

