# OpenReview forum: "Stochastic Concept Bottleneck Models"
_NeurIPS.cc/2024/Conference — NeurIPS 2024 poster_

### Official Review · Reviewer_9xQm · 2024-06-21

**Soundness:** 3
**Presentation:** 3
**Contribution:** 2
**Rating:** 5
**Confidence:** 4

**Summary:**

Focusing on concept bottleneck models, this paper extracts concept dependencies with a multivariate normal distribution and derives an intervention strategy based on the confidence region of the normal distribution that incorporates concept correlations for better interventional effectiveness.

**Strengths:**

1. This paper is organized and written well.
2. Concept bottleneck model is an important direction for xAI and even generalizability.

**Weaknesses:**

1. Experiment results show marginal improvement over existing methods.
2. The novelty and contribution of this paper is limited.

**Questions:**

1. Intervention by setting values can be difficult in some case, the example given by the paper: "intervene on the CBM by setting its value to 1" is an extreme case. Humans are not good at estimating probabilities.
2. "do not use the intervened-on concepts to update their remaining concept predictions" may not necessarily be a bad thing. As in many real-world cases, there are exceptions to rules and patterns. Nothing is absolute. Universally "extend the concept predictions with the modeling of their dependencies" may be problematic. Are these exceptions handled by confidence region?
3. What is the novelty comparing with E. Kim et al. (2023)? just relax the assumption of diagonal covariance matrix?
4. Will the proposed method suffer leakage?

**Limitations:**

Limitations are adequately addressed.

---

> ### Author Rebuttal · Authors · 2024-08-07
>
> We thank the reviewer for the feedback and questions! Below is our point-by-point response.
>
> > Intervention by setting values can be difficult in some case, the example given by the paper: "intervene on the CBM by setting its value to 1" is an extreme case. Humans are not good at estimating probabilities.
>
> Indeed, specifying an exact value for interventions is a challenging task and an active research problem [8]. Still, a major benefit of CBMs is their ability for interventions via human-model interactions [2,3], as it enables domain experts to correct mistakes, so that more accurate target predictions can be made at the end. This is important in domains like healthcare, where human-model interactions have a significant impact on trust and patient outcomes.
> Thus, we believe our research in this direction is important while acknowledging that the complementary problem of the human factor within interventions is not solved.
>
> Notably, while most CBM methods require users to intervene with exact values, SCBMs solve an optimization problem to arrive at a “reasonable” probability estimate. That is, not only do we not ask humans to specify probabilities, but we even solve an optimization problem to determine probabilities given hard concept values. Naturally, if desired by the user, this optimization routine can be omitted. Nevertheless, we will mention this problem in the limitation section.
>
> > "do not use the intervened-on concepts to update their remaining concept predictions" may not necessarily be a bad thing. As in many real-world cases, there are exceptions to rules and patterns. Nothing is absolute. Universally "extend the concept predictions with the modeling of their dependencies" may be problematic. Are these exceptions handled by confidence region?
>
> The goal of this work is to enhance the effect of individual interventions and make the human-model interactions more scalable. This removes the burden of intervening on multiple correlated concepts, leading to improved performance. SCBMs achieve this by exploiting the concept dependencies observed during training. As such, we do make the assumption that the concept structure in the training set holds at inference time. If this assumption is violated, one could always compute the marginal instead of the conditional distribution at intervention time, thereby ignoring the concept dependencies.
>
>
> > What is the novelty comparing with E. Kim et al. (2023)? just relax the assumption of diagonal covariance matrix?
>
> While both works use a normal distribution, we believe there are substantial differences between our manuscript and the ProbCBM [4], as we outline below. We will make them more clear in the revised version of this manuscript.
>
> * ProbCBMs build upon CEMs. That is, the mean and variance are learned over the concept *embeddings* rather than modeling the concepts themselves. Even if they relaxed the assumption of a diagonal covariance matrix, in their current form, they would only capture the covariances of the embedding dimensions *within* a given concept but not the dependencies *across* concepts. Thus, we would not expect that such a relaxation would improve ProbCBMs’ intervention performance, and as such, in the context of interventions, we do not consider them to be very different from CEMs.
>
> * In a similar vein, please note that the loss of ProbCBMs contains a KL term, which “prevents the variances from collapsing to zero” [4]. On the other hand, SCBMs do not require such a term, as we’re modeling actual non-zero concept dependencies. To draw an analogy, we consider ProbCBMs to work similarly to VAEs [5], where the normal distribution acts as a regularizing prior (via the KL term) on the concept latent space. On the other hand, SCBMs leverage the normal distribution to learn the structure of the data itself, which in our case, are the concepts.
>
> * As pointed out by the reviewer, a major difference between the works is that SCBMs do not assume a diagonal covariance matrix. However, this generalization is non-trivial and poses multiple technical challenges. As such, we parameterize the covariance via its Cholesky decomposition, derive a maximum-likelihood-based loss different from [4], apply the Gumbel-Softmax reparameterization to perform joint training, and introduce the Lasso regularization. Lastly, a significant contribution is the introduction of the novel intervention strategy based on the confidence region (see Eq. 8) that leverages the learned concept dependencies while fulfilling the posed desiderata.
>
> * While ProbCBMs mainly focus on capturing concept and target uncertainty, SCBMs focus on interventions. As we have shown in Fig. 2 of the manuscript, we achieve this gap by a margin of up to 10 percentage points in some cases. Especially when comparing to CEMs, whose performance we believe to be a good proxy for ProbCBMs with respect to interventions, there is a major difference in intervention effectiveness.
>
> * A seemingly minor but, in fact, very important difference is that ProbCBMs work with concept embeddings as introduced by CEMs. On the other hand, SCBMs use hard, binary concepts as bottleneck. We elaborate on the importance of this difference below.
>
> > Will the proposed method suffer leakage?
>
> Leakage is an important problem in the CBM literature, as it damages the interpretability. While we do not explicitly optimize against leakage, we have made the deliberate design choice to characterize the concept bottleneck with hard, binary concepts to avoid leakage as much as possible [6]. We believe that leakage is the main driver of why CEMs perform suboptimal during interventions. Note that [7] even propose to approximate leakage by performance during interventions. Since ProbCBMs build upon CEMs, we believe that they are more susceptible to leakage than SCBMs.
> But of course, our design choices do not fully prevent the occurrence of leakage, which is why we mention it in the limitations section.

---

> > ### Author Response · Authors · 2024-08-12
> >
> > Dear reviewer 9xQm,
> >
> > please let us know if we could address your concerns with our rebuttal or if there are other open questions remaining. We would be grateful if the reviewer could acknowledge the rebuttal. Thank you in advance!

---

> > > ### Comment · Reviewer_9xQm · 2024-08-13
> > >
> > > Thanks for the authors' response, which clarified some points.

---

### Official Review · Reviewer_sfUo · 2024-07-12

**Soundness:** 2
**Presentation:** 3
**Contribution:** 3
**Rating:** 5
**Confidence:** 4

**Summary:**

This paper introduces a novel concept dependency modelling scheme via an explicit distributional parameterization based on multivariate Gaussian distributions. This allows for capturing the dependencies between different concepts, while giving rise to an effective intervention strategy.

The experimental results vouch for the efficacy of the proposed approach, which allows for end-to-end training, and exhibits on-par or improved performance before and after interventions.

**Strengths:**

This constitutes a well-motivated approach based on some probabilistic arguments for capturing and examining  concept dependency. The overall idea is simple and easy to follow.

**Weaknesses:**

Even though the proposed approach uses either an amortized or global  formulation, the complexity of learning the covariance matrix either way surely introduces a lot of complexity. This is especially true in cases where the number of concepts are large, a common occurrence in complex datasets with many classes such as CIFAR-100 and ImageNet (not explored in this work). This complexity is further burdened by the Monte Carlo sampling scheme; commonly, 1-10 samples are enough, but the authors report the usage of 100 MC samples which would greatly slow down the training process.

In this context:
1) What is the complexity compared to a standard diagonal approach?
2) It would be important to assess the complexity and performance in the case of larger datasets like ImageNet. The authors use the CLIP generated data of [1] for CIFAR-10, so applying the approach to CIFAR-100 and ImageNet should be quite easy.
3) In my experience with MC-based methods, I find Table 2 a bit hard to believe, especially with 100 MC samples that the authors report. Can the authors provide wall time measurements for the per-epoch time for each method? Even for just Hard CBM and Amortized SCBM. 4) Having access to the code, so the experiments could be validated would also be important.
5) How does the number of MC samples affect the complexity and performance? I find an ablation study to be necessary here.
6) How many GPUs did the utilized cluster have?

The considered Bernoulli formulation is very similar to the work in [2], which they call concept discovery. It constitutes a data-driven Bernoulli posterior for concept selection and the formulation is very similar. I find this method more appropriate compared to other approaches mentioned in the related work and the experimental evaluation, and some discussion/results should be included in the main text.

In Table 1, the authors report the concept accuracy which I assume is based on a binary accuracy. Recent works have suggested that maybe this is not a good metric in the context of interpretability, since most concepts are sparse and this can lead to misleading results [3]. Can the authors report the Jaccard similarity between the ground truth and the obtained concepts for all the methods?

[1] Oikarinen, T., et al., Label-free concept bottleneck models, ICLR 2023

[2] Panousis, K. P., et al., Sparse Linear Concept Discovery Models, ICCV CLVL 2023

[3] Panousis, K. P., et al., Coarse-to-Fine Concept Bottleneck Models, Arxiv 2024

**Questions:**

Please see the Weaknesses section.

**Limitations:**

The authors mention the limitations in the dedicated section. The main limitation of the proposed approach is complexity.

---

> ### Author Rebuttal · Authors · 2024-08-07
>
> We thank the reviewer for their comments. In our response below, we address the remaining open points.
>
> > What is the complexity compared to a standard diagonal approach?
>
> As the reviewer rightly points out, modeling dependencies comes at a complexity overhead cost. In terms of memory complexity SCBMs scale quadratically with the number of concepts, while a diagonal approach would scale linearly. However, the sampling itself is very fast. We use the Cholesky decomposition $\Sigma = L L^T$; thus, sampling from the multivariate distribution is done via $Lx$, where x is a vector of $C$ samples of a univariate standard normal distribution. This is in stark contrast to the autoregressive baseline, which requires $C$ sequential passes through MLPs.
>
> > It would be important to assess the complexity and performance in larger datasets like ImageNet. The authors use the CLIP-generated data of [1] for CIFAR-10, so applying the approach to CIFAR-100 and ImageNet should be quite easy.
>
> We present the result on the CIFAR-100 dataset with 892 concepts obtained from [9] in Figure 1 of the rebuttal PDF to showcase the scalability of SCBMs. Additionally, we present the wall time per method in Table 2. As can be seen, the CEM and AR baselines take a long time to compute when increasing the concept set size. After 4 days, we are still waiting for the Autoregressive CBM to finish and can, unfortunately, only present their intervention performance up to this point. Therefore, we refrain from computing results on ImageNet with 4751 concepts, as we would be unable to get any results on time.
>
> The additional results underline the efficiency of our method in terms of computational complexity and performance. Notably, the Autoregressive baseline has a negative dip, which is likely due to the independently trained target predictor not being aligned with the concept predictors in this noisy CLIP-annotated scenario. Note that they need to train independently to avoid the sequential MC sampling during training, which would otherwise increase training time significantly, as seen in the test wall times in Table 2. Our jointly trained SCBMs do not have this issue and surpass the baselines.
>
> > … Can the authors provide wall time measurements for the per-epoch time for each method? Having access to the code, so the experiments could be validated would also be important.
>
> We chose 100 MC samples to not disadvantage the Autoregressive CBM, which used 200 MC samples. We provide an ablation for a smaller number of samples in Figure 2 of the supplementary PDF, showing SCBMs remain effective.
>
> In Table 2 of the supplementary PDF, we present the wall time for each method in two datasets with varying concept set sizes. Please note that the sampling itself is extremely fast for the multivariate Gaussian distribution used in SCBMs, and, contrary to e.g., VAEs, the MC samples are only passed through a lightweight target prediction head. This is a significant benefit over the autoregressive baseline, as their procedure consists of sequentially sampling each concept via MLPs, while SCBMs can obtain all samples directly from the specified multivariate Gaussian.
>
> Let us denote by $C$ the number of concepts and $M$ the number of MC samples.
> If we analyze the computational complexity by the amount of MLP forward passes, then CEMs scale $O(C)$ (2 embeddings per concept), Autoregressive CBMs scale $O(C \times M)$ (1 pass through $C$ MLPs per MC sample), and SCBMs scale $O(M)$ (1 pass through classifier head per MC sample).
>
> For code, we refer to the footnote on page 3 of the originally submitted manuscript for our anonymized repository.
>
> > How does the number of MC samples affect the complexity and performance?
>
> In Figure 2 of the supplementary PDF we provide an ablation for the number of MC samples on CUB. The runtimes differ minimally by ~0.01s per epoch. Results show that SCBMs still perform well with fewer MC samples. Still, a larger amount (e.g. 100) of samples is completely feasible in SCBMs, supported by our fast sampling and forward pass.
>
> > How many GPUs did the utilized cluster have?
>
> The experiments were run on an ordinary HPC platform, mostly consisting of GeForce RTX 2080ti’s. We would like to emphasize that each run uses only a single GPU.
> We will adjust the “Resource Usage” paragraph to avoid ambiguity.
>
> > The considered Bernoulli formulation is very similar to the work in [2] … I find this method more appropriate compared to other approaches mentioned in the related work and the experimental evaluation, and some discussion/results should be included in the main text.
>
> We agree that Panousis et al. (2023) and SCBMs share the Bernoulli relaxation for concept modeling. It is important to note that CDMs focus on discovering concepts, while SCBMs assume the presence of a concept-labeled dataset and focus on modeling their dependencies. As such, we believe the two works to be complementary, not substitutional. This is observable by noticing that both cited works by Panousis do not study interventions. We have reported successful results of SCBMs in CIFAR-10 and CIFAR-100, where concept annotations are “discovered” from CLIP. This shows that the concept discovery methods like CDMs can synergize with SCBMs. However, as the focus of this work lies on modeling the concept dependencies and interventions, we believe that autoregressive CBM is more closely related to our task and focus on it.
>
> We will make sure to discuss the shared similarities with CDMs in the main text, as well as outline the combination of both methods as an exciting avenue for future work.
>
> > In Table 1, the authors report the concept accuracy…. Can the authors report the Jaccard similarity?
>
> In Table 1 of the supplementary PDF, we include the Jaccard similarity between ground truth and predicted concepts. The resulting interpretations are in line with the prior reported accuracies. We will add this metric proposed by Panousis et al. (2024) to the manuscript.

---

> > ### Author Response · Authors · 2024-08-12
> >
> > Dear reviewer sfUo,
> >
> > please let us know if we could address your concerns with our rebuttal or if there are other open questions remaining. We would be grateful if the reviewer could acknowledge the rebuttal. Thank you in advance!

---

> > > ### Comment · Reviewer_sfUo · 2024-08-13
> > >
> > > I would like to thank the authors for their thorough rebuttal. I really appreciate the additional experiments using CIFAR-100 and the consideration of the proposed metric. ImageNet would also be important to evaluate at some point since it commonly considers a very large amount of concepts and it would be interesting to observe the complexity and the behavior of the proposed method.
> > >
> > > After the provided clarifications, I also agree that the proposed approach is complementary to Panousis et al. 2023 and I believe it would be a good probabilistic combination. To this end, and after reading the comments and the discussions with the other reviewers I decided to increase my score.

---

### Official Review · Reviewer_rLqE · 2024-07-14

**Soundness:** 4
**Presentation:** 3
**Contribution:** 4
**Rating:** 7
**Confidence:** 3

**Summary:**

This paper presents a method of performing interventions on Concept Bottleneck Models. The method parametrizes the space of concepts with a generative model of Bernoulli distribution and concept logits with a normal distribution, whose mean and variance depends on the input data distribution. The method also compares amortized covariance, which is instance-based, and a global covariance, marginalized over all samples. Rather than doing the interventions one-by-one as in previous works, this method also  modifies a single concept by hand, which in turns modifies other related concepts using the covariance learned. Experiments demonstrate that the method performs competitively against related CBM works that also allow for intervention, and is much faster during inference. During intervention, the model performs competitively against other baselines.

**Strengths:**

The problem presented in this work is well-motivated and is of great importance to the current field of interpretable machine learning. While there are existing works that also formulate the concepts as a known distribution, this method has demonstrated well its advantages from the lens of intervention. Hence, this work has met the bar in terms of novelty and contributions in this area of research. The experiments of this work are also solid and adequate in addressing the claims made. While the performance is sometimes worse than related methods, they are still comparable and is not the main focal point of this work.

**Weaknesses:**

While the overall writing of the paper is clear, Section 3.3 was a bit challenging to follow, especially the part about the confidence region. It is not entirely clear to me what the confidence region is of. As an example, L183-184 “the likelihood-based
confidence region provides a natural way of capturing the region of possible $\mathbf{\eta}’_{\mathcal{S}}$ that fulfil our desiderata”. The confidence of what? What desiderata? I suggesting either adding a diagram, a bit more background, a reminder, or an example, on what the intervention is trying to achieve so it more clearly demonstrates the process.

**Questions:**

Often times we don’t have many labeled samples of fine-grain annotations (like CUB) or the annotation can be noisy (like CLIP annotations). To what extend do these two factors affect the learning of the distributions in your method? Is it true that if the global dependency structure is of poor quality (correlation matrix), then this can severely impact the downstream performance and interpretability?

**Limitations:**

A key part of this method is learning a good dependency matrix for each sample or globally. I would expect this requires a lot of samples and can severely impact the performance if the matrix is not representative of the relationships between concepts. This is clearly stated in the work, hence the limitations are sufficiently stated.

---

> ### Author Rebuttal · Authors · 2024-08-07
>
> Dear reviewer,
> Thank you for your comments and positive feedback. Please find below our answers to the open points.
>
>
> > While the overall writing of the paper is clear, Section 3.3 was a bit challenging to follow, especially the part about the confidence region. It is not entirely clear to me what the confidence region is of. As an example, L183-184 “the likelihood-based confidence region provides a natural way of capturing the region of possible $\boldsymbol{\eta}_{\mathcal{S}}$ that fulfil our desiderata”. The confidence of what? What desiderata? I suggesting either adding a diagram, a bit more background, a reminder, or an example, on what the intervention is trying to achieve so it more clearly demonstrates the process.
>
> Thank you for raising this point. We will make sure to improve the clarity of this subsection in the manuscript.
> Below, we provide the rewritten version of lines 174-188.
>
> *In normal CBMs, an intervention affects only the concepts on which the user intervenes. As such, Koh et al. (2020) set $\eta_i'$ to the 5th percentile of the training distribution if $c_i = 0$ and the 95th percentile if $c_i = 1$. While this strategy is effective for SCBMs too, see Appendix C.3, the modeling of the concept dependencies warrants a more thorough analysis of the *intervention strategy*. We present two desiderata, which our intervention strategy should fulfill.*
>
> *i)  $p(c_i | \eta_i') \geq p(c_i | \mu_i)$*
>
> *That is, the likelihood of the intervened-on concept $c_i$ should always increase after the intervention. If SCBMs used the same strategy as CBMs, it could happen that the initially predicted $\mu_i$ was more extreme than the selected training percentile. Then, the interventional shift $\eta'_i - \mu_i$ in Equation 7 would point in the wrong direction. This, in turn, would cause $\boldsymbol{\eta}\_{\setminus \mathcal{S}}$ to shift incorrectly.*
>
> *ii) $|\eta_i' - \mu_i|$ should not be ``too large''*
>
> *We posit that the interventional shift should stay within a reasonable range of values. Otherwise, the effect on $\eta_{\setminus \mathcal{S}}$ would be unreasonably large such that the predicted $\boldsymbol{\mu}_{\setminus \mathcal{S}}$ would be completely disregarded.*
>
>
> *To fulfill these desiderata, we take advantage of the explicit distributional representation: the likelihood-based confidence region of $\mu_i$ provides a natural way of specifying the region of possible $\boldsymbol{\eta}\_{\mathcal{S}}'$ that fulfil our desiderata. Informally, a confidence region captures the region of plausible values for a parameter of a distribution.
> Note that the confidence region takes concept dependencies into account when describing the area of possible $\boldsymbol{\eta}\_{\mathcal{S}}'$. To determine the specific point within this region, we search for the values $\boldsymbol{\eta}\_{\mathcal{S}}'$, which maximize the log-likelihood of the known, intervened-on concepts $\mathbf{c}_{\mathcal{S}}$, implicitly focusing on concepts that the model predicts poorly.*
>
> > A key part of this method is learning a good dependency matrix for each sample or globally. Is it true that if the global dependency structure is of poor quality (correlation matrix), then this can severely impact the downstream performance and interpretability?
>
> Yes, modeling the second central moment naturally induces more variability. As the reviewer rightly states, it is important that the learned dependency structure is somewhat accurate. That is, the benefits of learning the dependencies have to outweigh the difficulty of learning them. This is why we introduced the LASSO-like regularizer, which helps in avoiding overfitting. For the regularizer weight $\lambda_2 \rightarrow \infty$, we would recover the hard CBM (apart from diagonal variance), thus, with a validation set at hand, one can find a suitable value. Note that we also show in App. C.2, that SCBMs are not very sensitive to this $\lambda_2$. The importance of not learning a poor-quality structure is also what led us to not model higher-order central moments, thereby reducing the overfitting potential.
>
>
>
>  > Often times we don’t have many labeled samples of fine-grain annotations (like CUB) or the annotation can be noisy (like CLIP annotations). To what extend do these two factors affect the learning of the distributions in your method?
>
> Learning a covariance structure adds a layer of complexity, therefore, it is natural to wonder how our method can deal with non-optimal scenarios. We believe with the regularization, discussed in the previous answer, users have good control over the method’s behavior and, therefore, good control over the potential issues you point out. Of course, noisier annotations will require a higher $\lambda_2$ regularization, thus, potentially omitting some signal. However, this tradeoff is unavoidable, and regularizing the parameters is standard practice [1]. Additionally, the proposed intervention strategy aids in preventing wrongful interventions by adhering to the predicted uncertainty.
>
> We have deliberately chosen the datasets in the manuscript to showcase the versatility of SCBMs in such settings. That is, the CUB dataset contains only around 6000 training samples, thus being a rather small dataset, in which SCBMs prevail. To cover the case of noisy concepts, as well as a dataset without fine-grained human annotations, we have chosen the CLIP-annotated CIFAR-10 dataset, in which SCBMs also outperform the baselines. These findings also hold out-of-the-box on the CLIP-annotated CIFAR-100 dataset reported in Figure 1 of the supplementary PDF file, showcasing the scalability of SCBMs to larger concept sets. Thus, we can confidently conclude that SCBMs are effective in capturing the underlying dependencies.

---

> > ### Author Response · Authors · 2024-08-12
> >
> > Dear reviewer rLqE,
> >
> > please let us know if we could address your concerns with our rebuttal or if there are other open questions remaining. We would be grateful if the reviewer could acknowledge the rebuttal. Thank you in advance!

---

> > ### Comment · Reviewer_rLqE · 2024-08-12
> >
> > Thank you for the author's detailed response. The improved writing improved my understanding of the work much better.

---

### Author Rebuttal · Authors · 2024-08-07

Dear reviewers,

We would like to thank all of you for your thorough reviews and constructive feedback! Below, we summarise our responses to your main concerns, additional results, and changes to be implemented upon acceptance in the revised manuscript.

* We have included experiments on a new large-scale dataset, CIFAR-100 [10], with annotations for 892 concepts coming from VLMs. The results, displayed in Figure 1 of the additional supplementary PDF file, are consistent with the interpretation of SCBMs in previous datasets, showcasing the scalability of our method to a larger number of concepts and classes. For a more detailed discussion, we refer to our response to reviewer sfUo.

* To understand the computational complexity of SCBM, we report the wall times for one training and inference epoch for each of the studied methods in Table 2 of the supplementary PDF. In view of the large-scale dataset introduced in the rebuttal, we report both CUB and CIFAR-100 to provide insights for two different datasets. These results show that SCBMs are significantly more efficient and hence more scalable than prior work for modeling concept dependencies, i.e., Autoregressive (AR), which required longer computational times, especially when scaling the number of concepts. Particularly, AR is slower at inference time, where speed is a key factor as a user will interact with the deployed model. For a more detailed discussion, we refer to our response to reviewer sfUo.

* The experiments in the proposed SCBM were carried out with 100 MCMC samples. We performed an ablation on this amount and show in Figure 2 of the supplementary PDF that SCBMs remain effective with 10 MCMC samples.

* We include the Jaccard Index as a new metric for measuring concept prediction prior to interventions for all models and datasets in Table 1 of the supplementary PDF. The resulting interpretation is consistent with the previously reported accuracies.

* We further clarified the method with a comprehensive explanation of the proposed intervention strategy via confidence regions, which we will include in the revised version of the manuscript. For more details, we refer to our response to reviewer rLqE.


[1] Tibshirani, Robert. "Regression shrinkage and selection via the lasso." Journal of the Royal Statistical Society Series B: Statistical Methodology 58.1 (1996): 267-288.

[2] Koh, Pang Wei, et al. "Concept bottleneck models." International conference on machine learning. PMLR, 2020.

[3] Shin, Sungbin, et al. "A closer look at the intervention procedure of concept bottleneck models." International Conference on Machine Learning. PMLR, 2023.

[4] Kim, Eunji, et al. "Probabilistic Concept Bottleneck Models." International Conference on Machine Learning. PMLR, 2023.

[5] Kingma, D. P., & Welling, M. “Auto-encoding Variational Bayes.” International Conference on Learning Representations, ICLR, 2014.

[6] Havasi, Marton, Sonali Parbhoo, and Finale Doshi-Velez. "Addressing leakage in concept bottleneck models." Advances in Neural Information Processing Systems 35 (2022): 23386-23397.

[7] Zabounidis, Renos, et al. "Benchmarking and Enhancing Disentanglement in Concept-Residual Models." arXiv preprint arXiv:2312.00192 (2023).

[8] Collins, Katherine Maeve, et al. "Human uncertainty in concept-based ai systems." Proceedings of the 2023 AAAI/ACM Conference on AI, Ethics, and Society. 2023.

[9] Oikarinen, T., et al., "Label-free concept bottleneck models", ICLR 2023

[10] Krizhevsky, A., & Hinton, G. (2009). “Learning multiple layers of features from tiny images.” Toronto, Ontario: University of Toronto.

---

### Decision · Program_Chairs · 2024-09-25

**Decision:**

Accept (poster)

**Comment:**

The paper introduces a novel Stochastic Concept Bottleneck Model (SCBM) that significantly enhances the functionality of Concept Bottleneck Models (CBMs) by modeling concept dependencies using a multivariate Gaussian distribution. This approach allows for more effective interventions where a change in one concept automatically adjusts related concepts, improving the model's overall interpretability and usability. Despite the complexity of learning the covariance matrix among concepts being a challenge, the authors effectively address this through detailed computational analyses and comparisons with baseline methods. They also commit to further clarifications in the final manuscript for sections that were difficult to understand, leading to a recommendation for acceptance.